# GM-VAE: Representation Learning with VAE on Gaussian Manifold

## Abstract

We propose a Gaussian manifold variational auto-encoder (GM-VAE) whose latent space consists of a set of diagonal Gaussian distributions. It is known that the set of the diagonal Gaussian distributions with the Fisher information metric forms a product hyperbolic space, which we call a Gaussian manifold. To learn the VAE endowed with the Gaussian manifold, we first propose a pseudo Gaussian manifold normal distribution based on the Kullback-Leibler divergence, a local approximation of the squared Fisher-Rao distance, to define a density over the latent space. With the newly proposed distribution, we introduce geometric transformations at the last and the first of the encoder and the decoder of VAE, respectively to help the transition between the Euclidean and Gaussian manifolds. Through the empirical experiments, we show competitive generalization performance of GM-VAE against other variants of hyperbolic- and Euclidean-VAEs. Our model achieves strong numerical stability, which is a common limitation reported with previous hyperbolic-VAEs.

## 1 Introduction

The geometry of latent space in generative models, such as the variational auto-encoders (VAE) (Kingma & Welling, 2013) and generative adversarial networks (GAN) (Goodfellow et al., 2020), reflects the structure of the representation of the data. Mathieu et al. (2019); Nagano et al. (2019); Cho et al. (2022) show that employing a hyperbolic space as the latent space improves in preserving the hierarchical structure of the data in the latent space. The expanded geometry is not just limited to the hyperbolic space, as the space can be other types of Riemannian manifolds, such as spherical manifolds (Xu & Durrett, 2018; Davidson et al., 2018) and the product of Riemannian manifolds with mixed curvatures (Skopek et al., 2019).

Meanwhile, it is known that univariate Gaussian distributions equipped with Fisher information metric (FIM) form a Riemannian manifold, sharing the manifold with Poincaré half-plane which is one of the four isometric hyperbolic models. This statistical manifold is known to have a metric tensor akin to that of the Poincaré half-plane (Costa et al., 2015), providing a possibility of viewing it as a hyperbolic space. Furthermore, the diagonal Gaussian distributions form a product of Riemannian manifolds showing the presence of an extended statistical manifold.

Based on the connection between hyperbolic spaces and statistical manifolds, in this work, we add an alternative perspective on hyperbolic VAEs with a viewpoint from the information geometry. Previously proposed hyperbolic VAEs rely on the distributions defined over the hyperbolic space. Riemannian normal and wrapped normal are commonly used as prior and variational distributions over the hyperbolic space. Unlike the Gaussian distribution in Euclidean space, these distributions suffer from numerical instability (Mathieu et al., 2019; Skopek et al., 2019). In addition, the Riemannian normal requires performing rejection sampling, which often generates too many unwanted samples.

From the information geometric perspective of the hyperbolic space, we introduce a new distribution, named a pseudo Gaussian manifold normal distribution (PGM normal). The Gaussian manifold, here, refers to the statistical manifold with univariate Gaussian distributions. The newly proposed distribution uses the KL divergence as a statistical distance between two distributions in the Gaussian manifold. Since the KL divergence approximates the squared Riemannian distance of the statistical manifold, derived from FIM, the proposed distribution follows the geometric property of the Gaussian

distributions. We show that the PGM normal is easy to sample, and the KL divergence between two PGM normals can be computed analytically.

With the PGM normal as prior and variational distributions, we define a Gaussian manifold VAE (GM-VAE), whose latent space is defined over the Gaussian manifold. Nevertheless, the data points are still assumed to be defined over the Euclidean space. To correct the mismatch between the data space and the latent space, we introduce a transformation from Euclidean to hyperbolic space at the last and the first layers of the encoder and decoder, respectively.

Empirical experiments with multiple datasets show that GM-VAE can achieve a competitive generalization performance against existing hyperbolic VAEs. During the experiments, we observe that the PGM normal is robust in terms of sampling and computation of the KL divergence, compared to the commonly-used hyperbolic distributions; we briefly explain the reason why others are numerically unstable. Analysis of the latent space exhibits that the geometrical structures and probabilistic semantics of the dataset can be captured in the representations learned with GM-VAE.

We summarize our contributions as follows:

- We propose a variant of VAE whose latent space is defined on a statistical manifold formed by diagonal Gaussian distributions.
- We propose a new distribution called pseudo Gaussian manifold normal distribution, which is easy to sample and has closed form KL-divergence, to train the VAE on the manifold.
- We propose new encoder and decoder structures to support the proper transition between Euclidean (data) space and the statistical manifold.
- We empirically verify that the newly proposed model performs similarly to existing hyperbolic VAEs while achieving stable training without numerical issues.

## 2 PRELIMINARIES

In this section, we first review the fundamental concepts of the Riemannian manifold. We then explain the commonly-used distributions over the Riemannian manifolds and visit the concepts of Riemannian geometry between statistical objects.

### 2.1 REVIEW OF RIEMANNIAN MANIFOLD

A $n$-dimensional Riemannian manifold consists of a manifold $\mathcal{M}$ and a metric tensor $g : \mathcal{M} \to \mathbb{R}^{n \times n}$, which is a smooth map from each point $\mathbf{x} \in \mathcal{M}$ to a symmetric positive definite matrix. The metric tensor $g(\mathbf{x})$ defines the inner product of two tangent vectors for each point of the manifold $\langle \cdot, \cdot \rangle_{\mathbf{x}} : \mathcal{T}_{\mathbf{x}} \mathcal{M} \times \mathcal{T}_{\mathbf{x}} \mathcal{M} \to \mathbb{R}$, where $\mathcal{T}_{\mathbf{x}} \mathcal{M}$ is the tangent space of $\mathbf{x}$.

A Riemannian manifold can be characterized by the curvature of the curves defined on it. The curvature of a Riemannian manifold can be computed at each point of the curves, while some manifolds have curvature of a constant value. For example, the unit sphere $\mathcal{S}$ has constant positive curvature of $+1$, and the Poincaré half-plane $\mathcal{U}$ has constant negative curvature of $-1$. The hyperbolic models Among the hyperbolic space, the Klein model, the Poincaré disk model, the Lorentz (Hyperboloid) model, and Poincaré half-plane model are known to be isometric and have the same value of curvature $-1$ (Nickel & Kiela, 2018; Gulcehre et al., 2018; Tifrea et al., 2018).

The metric tensor induces basic operations of the Riemannian manifold such as a geodesic, exponential map, log map, and parallel transport. Given two points $\mathbf{x}, \mathbf{y} \in \mathcal{M}$, geodesic $\gamma_{\mathbf{x}} : [0, 1] \to \mathcal{M}$ is a unit speed curve on $\mathcal{M}$ being the shortest path between $\gamma(0) = \mathbf{x}$ and $\gamma(1) = \mathbf{y}$. This can be interpreted as the generalized curve of a straight line in the Euclidean space. The exponential map $\exp_{\mathbf{x}} : \mathcal{T}_{\mathbf{x}} \mathcal{M} \to \mathcal{M}$ is defined as $\gamma(1)$, where $\gamma$ is a geodesic starting from $\mathbf{x}$ and $\gamma'(0) = \mathbf{v}$, where a tangent vector $\mathbf{v} \in \mathcal{T}_{\mathbf{x}} \mathcal{M}$. The log map $\log_{\mathbf{x}} : \mathcal{M} \to \mathcal{T}_{\mathbf{x}} \mathcal{M}$ is the inverse of the exponential map, i.e., $\log_{\mathbf{x}}(\exp_{\mathbf{x}}(\mathbf{v})) = \mathbf{v}$. The parallel transport $\mathrm{PT}_{\mathbf{x} \to \mathbf{y}} : \mathcal{T}_{\mathbf{x}} \mathcal{M} \to \mathcal{T}_{\mathbf{y}} \mathcal{M}$ moves the tangent vector $\mathbf{v}$ along the geodesic between $\mathbf{x}$ and $\mathbf{y}$. The distance function $d_{\mathcal{M}}(\mathbf{x}, \mathbf{y})$ can be induced from the metric tensor as follows:

$$d_{\mathcal{M}}(\mathbf{x}, \mathbf{y}) = \int_0^1 \sqrt{\langle \dot{\gamma}(t), \dot{\gamma}(t) \rangle_{\gamma(t)}} dt. \tag{1}$$

## 2.2 DISTRIBUTIONS OVER RIEMANNIAN MANIFOLD

Given a squared distance function $d_{\mathcal{M}}^2 : \mathcal{M} \times \mathcal{M} \to \mathbb{R}_{>0}$ of a Riemannian manifold $\mathcal{M}$, the probability density function of the Riemannian normal distribution can be computed by:

$$p_{\boldsymbol{\mu},\sigma}(\mathbf{z}) = \frac{1}{Z^{\mathcal{M}}} \exp \left( -\frac{d_{\mathcal{M}}^2(\mathbf{z}, \boldsymbol{\mu})}{2\sigma^2} \right), \tag{2}$$

where $\boldsymbol{\mu} \in \mathcal{M}$ is the Fréchet mean of the distribution, and $\sigma \in \mathbb{R}_{>0}$ is the dispersion parameter and $Z^{\mathcal{M}}$ is the normalizing factor. This is known to be preserving the maximum entropy property of the Gaussian distribution (Pennec, 2006). Note that the distribution requires computing the integral shown in Equation 1, which often does not have an analytic solution. In some special cases, one can compute the distance analytically but the computation is intractable in general. Mathieu et al. (2019) propose a rejection sampling method of the Riemannian normal defined on the Poincaré disk model, which we call a Poincaré normal distribution.

An alternative to Riemannian normal is the wrapped normal distribution. The wrapped normal distribution is constructed by transforming a sample from Gaussian distribution via parallel transportation and an exponential map:

$$\mathbf{z} = \exp_{\boldsymbol{\mu}}(\mathrm{PT}_{\mathbf{0}_{\mathcal{M}} \to \boldsymbol{\mu}}(f(\mathbf{v}))), \ \mathbf{v} \sim \mathcal{N}(\mathbf{0}, \Sigma), \tag{3}$$

where $\boldsymbol{\mu} \in \mathcal{M}$ is the mean vector of the distribution, $\mathbf{0}_{\mathcal{M}}$ is the origin of $\mathcal{M}$, $f(\cdot)$ maps a Euclidean vector to a tangent vector of $\mathbf{0}_{\mathcal{M}}$, and $\mathbf{v}$ is a sample obtained from Euclidean normal with the zero mean and covariance $\Sigma$. The probability density of the sample can be computed by using the change of variable technique. Note that $f(\cdot)$ is well-defined in hyperbolic spaces. For example, in the Lorentz model, we concatenate zero at the first dimension of the vector, and in the Poincaré disk model, it is an identity function. (Nagano et al., 2019) propose wrapped normal distribution on hyperbolic space, and we call it hyperbolic wrapped normal distribution.

## 2.3 STATISTICAL MANIFOLD

The parameter manifold $\mathcal{M}$ of the probability distributions $p_\theta : \mathcal{X} \to \mathbb{R}$, where $\theta \in \mathcal{M}$, equipped with the Fisher information metric (FIM) forms a Riemannian manifold (Rao, 1992). The FIM is defined as:

$$g_{ij}(\boldsymbol{\theta}) = \int_{\mathcal{X}} \frac{\partial \log p_\theta(x)}{\partial \theta_j} \frac{\partial \log p_\theta(x)}{\partial \theta_j} p_\theta(x) \, dx.$$

In the parameter space of univariate Gaussian distributions $\{(\mu, \sigma) \mid \mu \in \mathbb{R}, \sigma \in \mathbb{R}_{>0}\}$, the FIM can be simplified as two-dimensional diagonal matrix $\sigma^{-2}\mathrm{diag}(1, 0.5)$ (Costa et al., 2015). The diagonal form of the FIM implies that the Riemannian manifold with $\{(\mu, \sigma)\}$ has the same set of points as the manifold of the Poincaré half-plane, but with different curvature of value $-0.5$.

The parameter space of the $n$-dimensional diagonal Gaussian distributions becomes the product of $n$ manifolds of the parameter space of univariate Gaussian distributions. The operations on the product of the Riemannian manifolds $\bigotimes_{i=1}^n \mathcal{M}_i$ are defined manifold-wise. For example, an exponential map applied on a point $(p_i)_{i=1}^n \in \bigotimes_{i=1}^n \mathcal{M}_i$, with tangent vector $v_i \in \mathcal{T}_{p_i}\mathcal{M}_i$ for each $i \in \{1, \cdots, n\}$, can be represented as $(\exp_{p_i}(v_i))_{i=1}^n$.

## 2.4 STATISTICAL DISTANCE

The statistical distance is the distance, but may not be a metric, between two statistical objects such as random variables and probability density function. The statistical distance can provide similarities between two probability density functions.

On a statistical manifold equipped with FIM, a statistical distance called the Fisher-Rao distance can be well-derived. The Fisher-Rao distance of the statistical manifold is the Riemannian distance induced from the Fisher information metric using Equation 1. For example, the Fisher-Rao distance in the statistical manifold with the univariate Gaussian distribution can be easily induced using the Riemannian distance of the Poincaré half-plane model, where the Riemannian metric is similar (Costa et al., 2015).

Kullback-Leibler (KL) divergence is another widely-used statistical distance, which is defined as $D_{\mathrm{KL}}(p(x) \parallel q(x)) := \int_x p(x) \log \frac{p(x)}{q(x)} \, dx$ for two distributions $p(x), q(x)$ in the same statistical manifold. For example, the KL divergence for two univariate Gaussian distributions, $\mathcal{N}(\mu_1, \sigma_1)$ and $\mathcal{N}(\mu_2, \sigma_2)$, can be computed as:

$$D_{\mathrm{KL}}\left(\mathcal{N}(\mu_1, \sigma_1) \parallel \mathcal{N}(\mu_2, \sigma_2)\right) = \log \frac{\sigma_2}{\sigma_1} + \frac{\sigma_1^2 + (\mu_1 - \mu_2)^2}{2\sigma_2^2} - \frac{1}{2}.$$

For the $n$-dimensional diagonal Gaussians, the KL divergence is calculated by summing the KL divergence of the univariate Gaussians for each dimension. One notable property of KL divergence is that it can locally approximate the squared Fisher-Rao distance.

## 3 METHOD

In this section, we first derive a reparameterization of the Gaussian distribution to form a statistical manifold with an arbitrary curvature. We then propose a Pseudo Gaussian manifold (PGM) normal distribution. Finally, we suggest a new variant of the variational auto-encoder, whose latent space is defined over the statistical manifold.

### 3.1 MANIFOLD WITH ARBITRARY CURVATURE

As shown in Section 2.3, the univariate Gaussian distributions form a statistical manifold with a negative half curvature, whose manifold is the same as the manifold of Poincaré half-plane. Previous studies on hyperbolic spaces emphasize the importance of having an arbitrary curvature (Skopek et al., 2019; Mathieu et al., 2019). These works empirically show that the generalization performances of hyperbolic VAEs can be improved with varying curvatures.

We show that the statistical manifold of univariate Gaussian can have an arbitrary curvature by reparameterizing the Gaussian distribution properly. Let $\mathcal{N}(\sqrt{2c}\mu, \sigma)$ be the reparameterized Gaussian distribution with additional parameter $c > 0$. The reparameterization leads to the FIM of $\sigma^{-2}\mathrm{diag}(1, c)$ showing that the curvature of the statistical manifold is $-c$.

With the arbitrary curvature, we also verify that the KL divergence still approximates the Riemannian distance as:

$$\frac{D_{\mathrm{KL}}\left(\mathcal{N}(\sqrt{2c}(\mu + d\mu), \sigma + d\sigma) \parallel \mathcal{N}(\sqrt{2c}\mu, \sigma)\right)}{2c} = \frac{1}{2}\begin{pmatrix} d\mu \\ d\sigma \end{pmatrix}^T \begin{pmatrix} \frac{1}{\sigma^2} & 0 \\ 0 & \frac{1}{c\sigma^2} \end{pmatrix} \begin{pmatrix} d\mu \\ d\sigma \end{pmatrix} + \mathcal{O}\left((d\sigma)^3\right),$$
(4)

where the first term is the squared Riemannian norm of the vector $(d\mu, d\sigma)$ in the manifold, which approximates the squared Fisher-Rao distance between $(\mu, \sigma)$ and $(\mu + d\mu, \sigma + d\sigma)$. The derivation of FIM and KL divergence with the reparameterized normal is described in Appendix A.1 and A.2. We call the statistical manifold with Gaussian distributions having a curvature of $-c$ as the *Gaussian manifold* $\mathcal{G}_c$.

### 3.2 PSEUDO GAUSSIAN MANIFOLD NORMAL DISTRIBUTION

We propose a pseudo Gaussian manifold normal distribution (PGM normal) defining a distribution over the Gaussian manifold. Let $(\mu, \sigma) \in \mathcal{G}$ be a point in the Gaussian manifold. Inspired by the Riemannian normal, we define the probability density function of PGM normal distribution with KL-divergence as:

$$\mathcal{K}_c(\mu, \sigma; \alpha, \beta, \gamma^2) = \frac{(\sigma/\beta)^3}{Z(c, \gamma^2)} \exp\left(-\frac{D_{\mathrm{KL}}(\mathcal{N}(\sqrt{2c} \cdot \mu, \sigma) \parallel \mathcal{N}(\sqrt{2c} \cdot \alpha, \beta))}{(\sqrt{2c} \cdot \gamma)^2}\right),$$
(5)

where $\alpha, \beta$, and $\gamma^2$ are the parameters of the distribution, and $-c$ is the curvature. As shown in the previous section, the KL divergence approximate the Fisher-Rao distance from Gaussian distribution $\mathcal{N}(\sqrt{2c} \cdot \alpha, \beta)$ to $\mathcal{N}(\sqrt{2c} \cdot \mu, \sigma)$. Therefore, the PGM normal accounts for the geometric structure of the Guassian distributions.

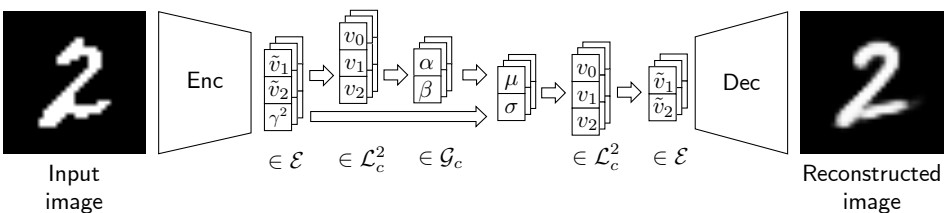

Figure 1: An examplar architecture of GM-VAE. The illustration shows the architecture of GM-VAE which sets the latent space to a three-dimensional diagonal Gaussian manifold. The encoder outputs parameters of the PGM normal, which are the points of the Gaussian manifold. The gray line refers to the sampling process. The decoder reconstructs data from the samples.

The factorization of the probability density function in Equation 5 multiplied with the square root of the determinant of the metric tensor shows the advantages of the PGM normal, which can be written as:

$$\mathcal{K}_c(\mu, \sigma; \alpha, \beta, \gamma^2) \cdot \sqrt{\det(g)} = \mathcal{N}(\mu; \alpha, \beta^2\gamma^2) \cdot \mathrm{Gamma}\left(\sigma^2; \frac{1}{4c\gamma^2} + 1, \frac{1}{4c\beta^2\gamma^2}\right), \quad (6)$$

where $\mathrm{Gamma}(z; a, b) = \frac{b^a}{\Gamma(a)} z^{a-1} \exp(-bz)$ and $g$ is the Fisher information metric of the Gaussian manifold. Note that the factorization has the same form as the well-known conjugate prior to the Gaussian distribution. In that sense, the PGM normal incorporates the geometric structure into the prior distribution explicitly. Thanks to the properties of Gaussian and Gamma distribution, the PGM normal is easy to sample and has a closed-form KL divergence. The detailed derivation is available in Appendix B.

### 3.3 GAUSSIAN MANIFOLD VAE

We propose a Gaussian manifold VAE (GM-VAE) whose latent space is defined over the Gaussian manifold. To be specific, we place a PGM normal prior over the latent space of the VAE and add a proper geometric transformation at the last layer of the encoder and the first layer of the decoder for the conversion between the Euclidean space and Gaussian manifold.

The evidence lower bound (ELBO) of the GM-VAE can be formalized with the Gaussian-manifold $\{(\boldsymbol{\mu}, \Sigma) \mid \boldsymbol{\mu} \in \mathbb{R}^n, \Sigma \in \mathbb{R}^n_{>0}\}$ as:

$$\mathbb{E}_{q_\phi(\boldsymbol{\mu}, \Sigma|\mathbf{x}) \cdot \sqrt{\det(g)}} \left[\log p_\theta(\mathbf{x} \mid \boldsymbol{\mu}, \Sigma)\right] - D_{\mathrm{KL}}\left(q_\phi(\boldsymbol{\mu}, \Sigma \mid \mathbf{x}) \cdot \sqrt{\det(g)} \| p(\boldsymbol{\mu}, \Sigma) \cdot \sqrt{\det(g)}\right), \quad (7)$$

where $p_\theta(\mathbf{x} \mid \boldsymbol{\mu}, \Sigma)$ is the decoder network, $q_\phi(\boldsymbol{\mu}, \Sigma \mid \mathbf{x})$ is the encoder network and $p(\boldsymbol{\mu}, \Sigma)$ is the prior. The variational distribution is set to $q_\phi(\boldsymbol{\mu}, \Sigma \mid \mathbf{x}) = \mathcal{K}(\alpha_\phi(\mathbf{x}), \beta_\phi(\mathbf{x}), \gamma^2_\phi(\mathbf{x}))$, where $\alpha_\theta(\mathbf{x}) \in \mathbb{R}^n$ and $\beta_\phi(\mathbf{x}), \gamma^2_\phi(\mathbf{x}) \in \mathbb{R}^n_{>0}$, and the prior is set to $p(\boldsymbol{\mu}, \Sigma) = \mathcal{K}(\mathbf{0}, I, I)$ in our experiments. The training for the parameters of GM-VAE ($\theta$ and $\phi$) is to maximize the ELBO.

---

**Algorithm 1** Encoder

**Input** Input data $\mathbf{x}$, Encoding layers $\mathrm{Enc}(\cdot)$

**Output** Parameter $(\alpha, \beta) \in \mathcal{G}_c, \gamma^2 \in \mathbb{R}_{>0}$

1: $\tilde{\mathbf{v}}, \gamma^2 = \mathrm{Enc}(\mathbf{x})$       $\triangleright \tilde{\mathbf{v}} \in \mathcal{E}$
2: $\mathbf{v} = \exp^c_{\mathbf{0}_\mathcal{L}}(f(\tilde{\mathbf{v}}))$       $\triangleright \mathbf{v} \in \mathcal{L}^2_c$
3: $(\alpha, \beta) = T_c(\mathbf{v})$       $\triangleright (\alpha, \beta) \in \mathcal{G}_c$
4: **return** $(\alpha, \beta), \gamma^2$

---

**Algorithm 2** Decoder

**Input** Sample $(\mu, \sigma) \sim \mathcal{K}(\cdot)$, Decoding layers $\mathrm{Dec}(\cdot)$

**Output** Reconstruction $\mathbf{x}'$

1: $\mathbf{v} = T^{-1}_c(\mu, \sigma)$       $\triangleright (\mu, \sigma) \in \mathcal{G}_c, \mathbf{v} \in \mathcal{L}^2_c$
2: $\tilde{\mathbf{v}} = \log^c_{\mathbf{0}_\mathcal{L}}(\mathbf{v})$       $\triangleright \tilde{\mathbf{v}} \in \mathcal{E}$
3: $\mathbf{x}' = \mathrm{Dec}(\mathbf{v})$
4: **return** $\mathbf{x}'$

---

**Geometric transformations on GM-VAE**   Mathieu et al. (2019) propose a transformation from a Euclidean space to the Poincaré disk to define a latent space over the Poincaré disk. We propose a novel transformation in VAE from a Euclidean space to the Gaussian manifold and vice versa.

For a numerically stable transformation between two spaces, we adopt operations defined on the Lorentz model, which is isometric to the half-plane manifold (Nickel & Kiela, 2018). The isometry $T_c : \mathcal{L}_c^2 \to \mathcal{G}_c$ between the two-dimensional Lorentz model with curvature $-c$ and the Gaussian manifold with curvature $-c$ can be defined as:

$$T_c((t, x, y)) = \left( \frac{-y}{\sqrt{c}(t - x)}, \frac{1}{\sqrt{c}(t - x)} \right),$$

and the inverse is as:

$$T_c^{-1}((x, y)) = \left( \frac{1 + cx^2 + y^2}{2\sqrt{c}y}, \frac{-1 + cx^2 + y^2}{2\sqrt{c}y}, -\frac{x}{y} \right).$$

In the encoder, we convert the output of the last layer, which is in the Euclidean space, to the Lorentz model using the exponential map at the origin and then convert it to the Gaussian manifold using $T_c$. In the decoder, we convert the input of the first layer, which is in the Gaussian manifold, to the Lorentz model using the inverse of the transformation $T_c^{-1}$ and then convert it to the Euclidean space using the log map at the origin of the Lorentz model. Figure 1 illustrates the architecture of GM-VAE and the pseudo code for the encoder and decoder are shown in Algorithm 1 and Algorithm 2.

**Remark**   Unlike a typical VAE, where the latent space consists of samples from a Gaussian distribution, the latent space of GM-VAE consists of a set of Gaussian distributions. With this aspect, GM-VAE can be considered as a hierarchical VAE with an additional prior over the Gaussian prior. However, instead of sampling another latent variable from the latent distribution, we directly transform the latent distribution itself to **x** in the decoder network via transformation from hyperbolic to Euclidean space. From this perspective, GM-VAE can also be considered as a variant of Poincaré VAE (Mathieu et al., 2019), whose latent space can be interpreted via the Gaussian manifold.

## 4 RELATED WORK

**Information geometry on VAE**   Focusing on the virtue of bridging probability theory and differential geometry, the adaptation of information geometry to the deep learning framework has been investigated in various aspects (Karakida et al., 2019; Bay & Sengupta, 2017; Gomes et al., 2022). Having said that, Han et al. (2020) show that the training process of VAE can be seen as minimizing the distance between the two statistical manifolds: manifolds with the parameters of the decoder and the encoder. Not only can the parameters but the outputs from the VAE decoder be modeled as probability distributions. Arvanitidis et al. (2021) suggest a method of using the pull-back metric defined with arbitrary decoders on latent space. Our work focuses more on the statistical manifolds lying on the outputs of the encoder with the benefits from the information geometry.

**VAE with Riemannian manifold latent space**   The latent space of VAE reflects the geometrical property of the representations of the data. The efficacy of setting the latent space to be hyperbolic space (Mathieu et al., 2019; Nagano et al., 2019; Cho et al., 2022) or spherical space (Xu & Durrett, 2018; Davidson et al., 2018) has been verified for various datasets. Skopek et al. (2019) further extends the approach to enable the latent space to be the product of Riemannian manifolds with

Table 1: A comparison of the PGM normal (ours) with the commonly-used distributions on the hyperbolic space: Hyperbolic wrapped normal and Poincaré normal. Our method enables the easy sampling and computation of closed-form KL, with the utilization of the information geometry.

|  | Easy sampling | Information geometry | Closed-form KL |
|---|---|---|---|
| Hyperbolic wrapped normal | ◯ | × | × |
| Poincaré normal | △ | ◯ | × |
| PGM normal (ours) | ◯ | ◯ | ◯ |

different learnable curvatures. On top of these arts, we explore the method of setting the latent space to be a diagonal Gaussian manifold, which is isometric to the product of the hyperbolic space, providing a novel viewpoint on prior work with information geometry.

**Distributions on the hyperbolic space**    Defining a distribution in the hyperbolic space with easy sampling is challenging. Nagano et al. (2019) suggests hyperbolic wrapped normal distribution from the observation that the tangent space is Euclidean space. Leveraging operations defined on the tangent spaces, e.g., parallel transport, enables an easy sampling algorithm. Mathieu et al. (2019) propose a sampling method for the Riemannian normal defined on the Poincaré disk model using rejection sampling. This method rejects the pathological samples and enables accurate sampling from the distribution, but this demands a high amount of time complexity. These distributions are applied in many cases (Cho et al., 2022; Skopek et al., 2019; Mathieu & Nickel, 2020) but suffer from stability issues because of the absence of closed-form KL divergence. Our proposed distribution, however, not only share the common merits but also has overcome the stability problem with closed-form KL divergence. Table 1 summarizes the properties of each distribution.

## 5 EXPERIMENTS

In this section, we compare the performance of GM-VAE with the three baselines: Euclidean VAE, hyperbolic wrapped normal VAE (HWN VAE), and Poincaré VAE. The Euclidean VAE is the standard VAE with Euclidean latent space. The HWN VAE uses the product of two-dimensional Lorentz models as a latent space and uses the hyperbolic wrapped normal to model the prior and variational distributions. The Poincaré VAE uses the product of two-dimensional Poincaré disk models as a latent space and uses the Poincaré normal to model the prior and variational distributions. The Euclidean VAE, HWN VAE, and Poincaré VAE are denoted as $\mathcal{E}$-VAE, $\mathcal{L}$-VAE, $\mathcal{P}$-VAE, respectively in the following results.

### 5.1 DENSITY ESTIMATION

We first conduct a density estimation task to check the generalization ability of different models. We use three datasets: binarized-MNIST (Deng, 2012), binarized-Omniglot (Lake et al., 2015), and the images from Atari 2600 Breakout with binarization (binarized-Breakout) (Nagano et al., 2019). The binarized-Breakout are collected from plays with a pre-trained Deep Q-Network (Mnih et al., 2015). The size of images are $28 \times 28$, $28 \times 28$, and $80 \times 80$ for binarized-MNIST, binarized-Omniglot, and binarized-Breakout, respectively. The value of the threshold for binarization is set to 0.5, 0.5, and 0.1 for binarized-MNIST, binarized-Omniglot, and binarized-Breakout, respectively; the threshold

Table 2: Density estimation on real-world datasets. $d$ denotes the latent dimension. We report the negative test log-likelihoods of average 10 runs for binarized-MNIST and binarized-Omniglot, and an average 5 runs for binarized-Breakout with the 95% confidence interval. N/A in the log-likelihood indicates that the results are not available due to the failure of all runs, and N/A in the standard deviation indicates the results are not available due to failures of some runs. The best results are bolded.

| | $d$ | $\mathcal{E}$-VAE | $\mathcal{L}$-VAE | $\mathcal{P}$-VAE | GM-VAE $(c=1)$ | GM-VAE $(c=1/2)$ | GM-VAE $(c=3/2)$ |
|---|---|---|---|---|---|---|---|
| MNIST | 10 | $\mathbf{79.60_{\pm.13}}$ | $79.95_{\pm.19}$ | $80.52_{\pm.20}$ | $80.34_{\pm.30}$ | $80.10_{\pm.20}$ | $80.38_{\pm.18}$ |
| | 20 | $74.48_{\pm.46}$ | $73.67_{\pm.32}$ | $\mathbf{72.95_{\pm.11}}$ | $73.27_{\pm.22}$ | $73.31_{\pm.29}$ | $73.28_{\pm.19}$ |
| | 30 | $73.80_{\pm.07}$ | $73.46_{\pm.23}$ | $\mathbf{72.94_{\pm.10}}$ | $73.49_{\pm.13}$ | $73.35_{\pm.16}$ | $73.55_{\pm.22}$ |
| Omniglot | 10 | $136.53_{\pm.30}$ | $136.25_{\pm.36}$ | $134.95_{\pm.47}$ | $134.01_{\pm.28}$ | $135.20_{\pm.21}$ | $\mathbf{133.79_{\pm.30}}$ |
| | 20 | $121.18_{\pm.33}$ | $119.95_{\pm.40}$ | $\mathbf{117.79_{\pm.13}}$ | $118.79_{\pm.53}$ | $118.73_{\pm.39}$ | $119.03_{\pm.57}$ |
| | 30 | $118.67_{\pm.67}$ | $117.16_{\pm.48}$ | $\mathbf{115.09_{\pm.56}}$ | $117.97_{\pm.35}$ | $117.70_{\pm.47}$ | $117.95_{\pm.37}$ |
| Breakout | 24 | $50.37_{\pm.46}$ | $50.82_{\text{N/A}}$ | N/A | $\mathbf{49.35_{\pm.67}}$ | $50.82_{\pm.92}$ | $49.88_{\pm.51}$ |
| | 28 | $48.07_{\pm.20}$ | $48.74_{\text{N/A}}$ | N/A | $47.01_{\pm.31}$ | $48.31_{\pm.47}$ | $\mathbf{46.82_{\pm.77}}$ |
| | 32 | $48.06_{\pm.18}$ | $48.64_{\text{N/A}}$ | N/A | $47.04_{\pm.23}$ | $48.13_{\pm.30}$ | $\mathbf{46.88_{\pm.25}}$ |

Table 3: Ablation study on the geometric transformations of GM-VAE. Vanilla denotes the models without the geometric transformations, and Geo denotes the models with the geometric transformations. The geometric transformations enhance the generalization performance in most cases.

|  | $d$ | $c = 1$ Vanilla | Geo | $c = 1/2$ Vanilla | Geo | $c = 3/2$ Vanilla | Geo |
|---|---|---|---|---|---|---|---|
| MNIST | 10 | $80.28_{\pm.17}$ | $80.34_{\pm.30}$ | $80.28_{\pm.19}$ | $80.10_{\pm.20}$ | $80.38_{\pm.20}$ | $80.38_{\pm.18}$ |
|  | 20 | $75.33_{\pm.75}$ | $73.27_{\pm.22}$ | $74.94_{\pm.58}$ | $73.31_{\pm.29}$ | $75.74_{\pm.70}$ | $73.28_{\pm.19}$ |
|  | 30 | $74.63_{\pm.08}$ | $73.49_{\pm.13}$ | $74.48_{\pm.50}$ | $73.35_{\pm.16}$ | $75.05_{\pm.43}$ | $73.55_{\pm.22}$ |
| Omniglot | 10 | $135.12_{\pm.35}$ | $134.01_{\pm.28}$ | $135.47_{\pm.24}$ | $135.20_{\pm.21}$ | $134.69_{\pm.37}$ | $133.79_{\pm.30}$ |
|  | 20 | $122.24_{\pm.67}$ | $118.79_{\pm.53}$ | $121.10_{\pm.59}$ | $118.73_{\pm.39}$ | $122.44_{\pm.61}$ | $119.03_{\pm.57}$ |
|  | 30 | $120.85_{\pm.33}$ | $117.97_{\pm.35}$ | $119.48_{\pm.56}$ | $117.70_{\pm.47}$ | $121.23_{\pm.68}$ | $117.95_{\pm.37}$ |
| Breakout | 24 | $51.69_{\pm.20}$ | $49.35_{\pm.67}$ | $51.04_{\pm.72}$ | $50.82_{\pm.92}$ | $51.57_{\pm.50}$ | $49.88_{\pm.51}$ |
|  | 28 | $49.56_{\pm.69}$ | $47.01_{\pm.31}$ | $49.51_{\pm.51}$ | $48.31_{\pm.47}$ | $49.45_{\pm.63}$ | $46.82_{\pm.77}$ |
|  | 32 | $49.45_{\pm.29}$ | $47.04_{\pm.23}$ | $49.03_{\pm.38}$ | $48.13_{\pm.30}$ | $49.07_{\pm.54}$ | $46.88_{\pm.25}$ |

for binarized-Breakout is determined to visualize the components clear. The other details on the implementation and experimental setups are described in Appendix D.

The results are reported at Table 2. In binarized-MNIST and binarized-Omniglot, the models learned on the product hyperbolic space and the Gaussian manifold mostly outperform the Euclidean VAE. In binarized-Breakout, the GM-VAE with curvature values 1 and 3/2 outperform the baselines while the Poincaré VAE fails to run in all the settings and the HWN VAE fails to run in some of the settings due to numerical issues, which we further investigate in details.

**Numerical stability** We conduct an analysis of the numerical stability of the PGM normal distribution compared to the HWN and Poincaré normal. During the density estimation experiment, the HWN VAE and Poincaré VAE are often shown to be numerically unstable and fail to run in binarized-Breakout. Similar observations have been reported in several previous works (Mathieu et al., 2019; Chen et al., 2021; Skopek et al., 2019).

The hyperbolic wrapped normal uses the exponential map when transforming the output of the encoder to the Lorentz model and during the sampling, as described in Equation 3. The overlapped Lorentz model exponential map often causes an overflow. In the training of Poincaré VAE, the KL divergence between the variational distribution and the prior distribution needs to be approximated by the log-probability of the samples due to the absence of closed-form KL divergence in Poincaré normal. To compute the log probability of a given sample, the distance between two Poincaré disk model points, which are the sample and the Fréchet mean of the distribution, needs to be calculated, where the denominator term is numerically unstable. The PGM normal, on the other hand, is free

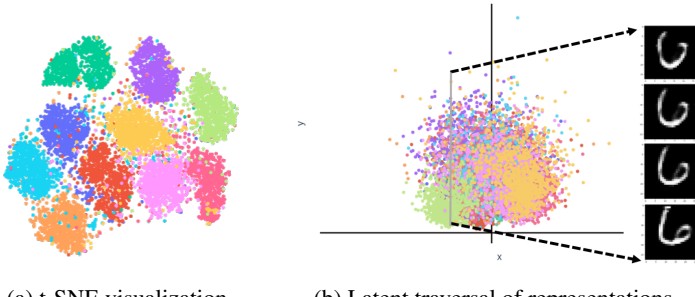

(a) t-SNE visualization.      (b) Latent traversal of representations.

Figure 2: Analysis of the learned latent space of GM-VAE with binarized-MNIST. (a) t-SNE visualization of the representation with respect to the class labels. (b) Increasing the value of $\beta$, along the gray line, results in an increasing degree of uncertainty in the reconstructed images.

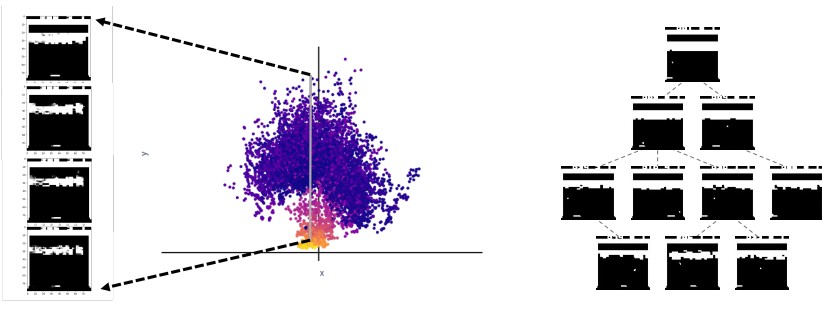

(a) Latent traversal of representations.          (b) Hierarhcy in the data.

Figure 3: Analysis of the latent space learned from GM-VAE with the binarized-Breakout. (a) Reconstructing the representations, along the gray line, shows a similar hierarchical structure, where (b) the hierarchy between the images is expressed as the dotted line.

from instability with the help of stability when using log-covariance. Please check the detailed arguments with equations in Appendix E.

**Geometric transformations**   We conduct an ablation study on the geometric transformations of GM-VAE. We compare the setting of GM-VAEs incorporating the geometric transformations to the setting of GM-VAEs using only exponential function to send the output of the encoder to the Gaussian manifold but no additional geometric transformation at the first layer of the decoder. The results are in Table 3. We can see that the geometric transformations enhance the performance of the GM-VAE, except for two results but with similar performance.

## 5.2   LATENT SPACE ANALYSIS

To check whether the latent representation coincides with the known labels, we first plot the latent spaces of binarized-MNIST via t-SNE (van der Maaten & Hinton, 2008) visualization, with representations from all dimensions. The visualization shown in Figure 2a presents that the label semantics are well clustered in the learned latent space. We also analyze the changes in the reconstructed images along the geodesic of the latent space. Figure 2b shows the reconstructed images from a geodesic interpolation between two latent representations, with a fixed value of $\alpha$. The interpolation of the latent space is performed within one dimension while fixing the value of representations in other dimensions. As $\beta$ increases, the reconstructed images become ambiguous, matching our intuition on the role of variance. Reconstruction images with a fixed value of $\beta$ is available at Appendix F.

Figure 3 shows the analysis with binarized-Breakout. The images in the binarized-Breakout possess a hierarchy as the cumulative rewards and the amount of the breakout bricks, or the portion of blank space in the image, are highly correlated (Nagano et al., 2019). We observe that there is a high correlation between $\beta$ and the hierarchy. For example, as shown in Figure 3a, increasing $\beta$ reconstructs a more general image in the hierarchy. The highest Pearson correlation between the $\beta$ values and the negative cumulative reward is 0.655. Again, as $\beta$ represents the variance, we conjecture the increasing variance induces a more general image in the dataset.

## 6   CONCLUSION

In this work, we propose a novel method of representation learning with GM-VAE, utilizing the Gaussian manifold for the latent space. With the newly-proposed PGM normal distribution defined over Gaussian manifold, which shows better stability and ease of sampling compared to the commonly-used ones, we verify the efficacy of our method on several real-world datasets. Our analysis of latent space and representations exhibits that GM-VAE is beneficial for capturing both the geometrical structures and probabilistic semantics. We believe that the connection between the statistical manifold and hyperbolic spaces provides a new insight to the research community and hope to see more interesting connections and analyses in the future.

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

# A  GAUSSIAN MANIFOLD

## A.1  CURVATURE OF THE GAUSSIAN MANIFOLD

We construct a Riemannian manifold $\{(\mu, \sigma) \mid \mu \in \mathbb{R}, \sigma \in \mathbb{R}_{>0}\}$ with the metric tensor $\mathrm{diag}(1/\sigma^2, 1/(c\sigma^2))$, which we will name Gaussian manifold. We need to show the value of the curvature.

First, we need to compute the Christoeffel symbols of the Gaussian manifold defined as:

$$\Gamma^k_{ij} = \frac{1}{2} g^{kl} \left( \frac{\partial g_{jl}}{\partial g_i} + \frac{\partial g_{il}}{\partial g_j} - \frac{\partial g_{ij}}{\partial g_l} \right), \tag{8}$$

where $g_{ij}$ is the $(i, j)$ element of the metric tensor and $g^{ij}$ is the $(i, j)$ element of the inverse of the metric tensor.

The Christoeffel symbols of the Gaussian manifold are:

$$\Gamma^1_{ij} = \begin{pmatrix} 0 & -\frac{1}{\sigma} \\ -\frac{1}{\sigma} & 0 \end{pmatrix} \tag{9}$$

$$\Gamma^2_{ij} = \begin{pmatrix} \frac{c}{\sigma} & 0 \\ 0 & -\frac{1}{\sigma} \end{pmatrix}. \tag{10}$$

Then, the sectional curvature of the space $\kappa_g$ is computed as:

$$\kappa_g = \frac{Rm(\mu, \sigma, \sigma, \mu)}{\det g}, \tag{11}$$

where $Rm$ is the Riemannian curvature which is computed as:

$$Rm(\mu, t, t, \mu) = g_{1m} \left( \frac{\partial \Gamma^m_{22}}{\partial \mu} - \frac{\partial \Gamma^m_{12}}{\partial \sigma} + \Gamma^p_{22} \Gamma^m_{1p} - \Gamma^p_{12} \Gamma^m_{2p} \right). \tag{12}$$

By putting the metric tensor and the Christoeffel symbols together, the curvature of the Gaussian manifold is computed as:

$$\kappa_g = \frac{Rm(\mu, \sigma, \sigma, \mu)}{\det g} = \frac{-\frac{1}{\sigma^4}}{\frac{1}{c\sigma^4}} = -c. \tag{13}$$

### A.2 GAUSSIAN MANIFOLD WITH KL-DIVERGENCE

Between two univariate Gaussian distributions $\mathcal{N}(\mu_1, \sigma_1^2)$ and $\mathcal{N}(\mu_2, \sigma_2^2)$, we can compute the KL divergence as:

$$D_{\mathrm{KL}}(\mathcal{N}(\mu_1, \sigma_1) \parallel \mathcal{N}(\mu_2, \sigma_2)) = \frac{1}{2} \left( \log \frac{\sigma_2^2}{\sigma_1^2} + \frac{\sigma_1^2 + (\mu_1 - \mu_2)^2}{\sigma_2^2} - 1 \right). \tag{14}$$

We extend the KL divergence for an arbitrary curvature of the Gaussian manifold as:

$$\mathcal{G}_{\mathrm{KL}}^c((\mu_1, \sigma_1), (\mu_2, \sigma_2)) := \frac{D_{\mathrm{KL}}(\mathcal{N}(\sqrt{2c}\mu_1, \sigma_1) \parallel \mathcal{N}(\sqrt{2c}\mu_2, \sigma_2))}{2c}. \tag{15}$$

Now, we show that the extended KL divergence still approximates the Riemannian distance of the manifold as:

$$\mathcal{G}_{\mathrm{KL}}^c((\mu + d\mu, \sigma + d\sigma), (\mu, \sigma)) = \frac{1}{2 * 2c} \left( \log \frac{\sigma^2}{(\sigma + d\sigma)^2} + \frac{(\sigma + d\sigma)^2 + 2c(d\mu)^2}{\sigma^2} - 1 \right) \tag{16}$$

$$= \frac{1}{2 * 2c} \left( -2 \log \left( 1 + \frac{d\sigma}{\sigma} \right) + \frac{2\sigma d\sigma + (d\sigma)^2}{\sigma^2} + \frac{2c(d\mu)^2}{\sigma^2} \right) \tag{17}$$

$$= \frac{1}{2 * 2c} \left( -2 \left( \frac{d\sigma}{\sigma} - \frac{(d\sigma)^2}{2\sigma^2} \right) + \frac{2\sigma d\sigma + (d\sigma)^2}{\sigma^2} + \frac{2c(d\mu)^2}{\sigma^2} + \mathcal{O}((d\sigma)^3) \right) \tag{18}$$

$$= \frac{1}{2} \begin{pmatrix} d\mu \\ d\sigma \end{pmatrix}^T \begin{pmatrix} \frac{1}{\sigma} & 0 \\ 0 & \frac{1}{c\sigma^2} \end{pmatrix} \begin{pmatrix} d\mu \\ d\sigma \end{pmatrix} + \mathcal{O}((d\sigma)^3). \tag{19}$$

## B PSEUDO GAUSSIAN MANIFOLD NORMAL DISTRIBUTION

In this section, we propose a pseudo-Gaussian-manifold normal distribution for the Gaussian manifold as:

$$\mathcal{K}_c(\mu, \sigma; \alpha, \beta, \gamma^2) = \frac{\sigma^3}{Z(\gamma) \cdot \beta^3} \exp \left( -\frac{\mathcal{G}_{\mathrm{KL}}^c((\mu, \sigma), (\alpha, \beta))}{\gamma^2} \right). \tag{20}$$

The given probability density function needs to satisfies the following condition:

$$\int_{\mathcal{G}} \mathcal{K}_c(\mu, \sigma; \alpha, \beta, \gamma^2) \sqrt{|g(\mu, \sigma)|} d(\mu, \sigma), \tag{21}$$

where $\sqrt{|(\mu, \sigma)|} d(\mu, \sigma)$ is the probability measure over the Gaussian manifold induced with the Lebesgue measure $d(\mu, \sigma)$ and the Lebesgue-Radon-Nikodym theorem. We can find the normalizing

factor $Z(\gamma)$ as:

$$
\begin{aligned}
Z(\gamma) &= \int_0^\infty \int_{-\infty}^\infty \left(\frac{\sigma}{\beta}\right)^3 \cdot \exp\left(-\frac{\mathcal{G}_{\mathrm{KL}}^c((\mu,\sigma),(\alpha,\beta))}{\gamma^2}\right) \frac{1}{\sqrt{c}\sigma^2}\, d\mu\, d\sigma \\
&= \frac{1}{\sqrt{c}\beta^3} \int_0^\infty \int_{-\infty}^\infty \sigma \cdot \exp\left(-\frac{\mathcal{G}_{\mathrm{KL}}^c((\mu,\sigma),(\alpha,\beta))}{\gamma^2}\right) d\mu\, d\sigma \\
&= \frac{1}{\sqrt{c}\beta^3} \left(\beta^{-\frac{1}{-2c\gamma^2}} \exp\left(\frac{1}{4c\gamma^2}\right) \int_0^\infty \sigma \cdot (\sigma^2)^{\left(\frac{1}{4c\gamma^2}+1\right)-1} \exp\left(-\frac{\sigma^2}{4c\beta^2\gamma^2}\right) d\sigma\right) \\
&\qquad\qquad\qquad\qquad\qquad\qquad \times \left(\int_{-\infty}^\infty \exp\left(-\frac{(\mu-\alpha)^2}{2\beta^2\gamma^2}\right) d\mu\right) \\
&= \frac{1}{2\sqrt{c}\beta^3}\sqrt{2\pi}\beta^3\gamma \exp\left(\frac{1}{4c\gamma^2}\right) \Gamma\left(\frac{1}{4c\gamma^2}\right)\left(\frac{1}{4c\gamma^2}\right)^{-\frac{1}{4c\gamma^2}} \\
&\qquad\qquad \times \left(\int_0^\infty \mathrm{Gamma}\left(\sigma^2;\frac{1}{4c\gamma^2}+1,\frac{1}{4c\beta^2\gamma^2}\right) d\sigma^2\right)\left(\int_{-\infty}^\infty \mathcal{N}(\mu;\alpha,\beta^2\gamma^2)\, d\mu\right) \\
&= \frac{\sqrt{2\pi}}{2\sqrt{c}}\gamma \exp\left(\frac{1}{4c\gamma^2}\right) \Gamma\left(\frac{1}{4c\gamma^2}\right)\left(\frac{1}{4c\gamma^2}\right)^{-\frac{1}{4c\gamma^2}}.
\end{aligned}
$$

Finally, the logarithm of the normalizing factor is computed as:

$$
\log Z(\gamma) = \frac{1}{2}\log(2\pi) - \frac{1}{2}\log c - \log 2 + \frac{1}{2}\log\gamma^2 + \log\Gamma\left(\frac{1}{4c\gamma^2}\right) + \frac{1}{4c\gamma^2}(1+\log(4c\gamma^2)). \tag{22}
$$

## C   OPERATIONS OF THE HYPERBOLIC SPACES

### C.1   ISOMETRIES

In this section, we derive the isometries between the two-dimensional hyperbolic models, the Lorentz model, the Poincaré disk model and the Gaussian manifold with arbitrary curvatures. Isometry between the Poincaré disk model and the Lorentz model $T_{\mathcal{L}_c \to \mathcal{P}_c} : \mathcal{L}_c \to \mathcal{P}_c$ is computed as:

$$
T_{\mathcal{L}_c \to \mathcal{P}_c}((t,x,y)) = \left(\frac{x}{\sqrt{c}t+1}, \frac{y}{\sqrt{c}t+1}\right),
$$

and the inverse is:

$$
T_{\mathcal{L}_c \to \mathcal{P}_c}^{-1}((x,y))\left(\frac{1+(x^2+y^2)c}{\sqrt{c}(1-(x^2+y^2)c)}, \frac{2x}{1-(x^2+y^2)c}, \frac{2y}{1-(x^2+y^2)c}\right). \tag{23}
$$

Isometry between the Gaussian manifold and the Poincaré disk model $T_{\mathcal{P}_c \to \mathcal{U}_c} : \mathcal{P}_c \to \mathcal{U}_c$ is computed as:

$$
T_{\mathcal{P}_c \to \mathcal{G}_c}(x,y) = \left(\frac{-2y}{(\sqrt{c}x-1)^2 + y^-2c}, \frac{1-(x^2+y^2)c}{(\sqrt{c}x-1)^2 + y^-2c}\right),
$$

and the inverse is:

$$
T_{\mathcal{P}_c \to \mathcal{G}_c}^{-1}(x,y)\left(\frac{\sqrt{c}x^2 + (y^2-1)/\sqrt{c}}{cx^2 + (y+1)^2}, \frac{-2x}{cx^2 + (y+1)^2}\right). \tag{24}
$$

Finally, the isometry between the Gaussian manifold and the Lorentz model $T_{\mathcal{L}_c \to \mathcal{G}_c}$ can be derived by composing $T_{\mathcal{L}_c \to \mathcal{P}_c}$ and $T_{\mathcal{P}_c \to \mathcal{G}_c}$ as:

$$
T_{\mathcal{L}_c \to \mathcal{G}_c}(t,x,y) = \left(\frac{-y}{\sqrt{c}(t-x)}, \frac{1}{\sqrt{c}(t-x)}\right),
$$

and the inverse is:

$$
T_{\mathcal{L}_c \to \mathcal{G}_c}^{-1}(x,y) = \left(\frac{1+cx^2+y^2}{2\sqrt{c}y}, \frac{-1+cx^2+y^2}{2\sqrt{c}y}, -\frac{x}{y}\right). \tag{25}
$$

We then empirically show that the isometries preserve the distance between the points when transformed to other models. We randomly sampled 1,000 pairs of Gaussian manifold points with range of $\mu \in [-100, 100]$ and $\sigma \in [0, 100]$. We report the average difference in the distance for each pair before the transformation and after the transformation. We vary the curvature value from 0.25 to 2. For the Gaussian manifold, we use the following distance function for arbitrary curvature:

$$d_{\mathcal{U}_c}((x_1, y_1), (x_2, y_2)) = \frac{1}{\sqrt{c}} \log \frac{\sqrt{c(x_1 - x_2)^2 + (y_1 + y_2)^2} + \sqrt{c(x_1 - x_2)^2 + (y_1 - y_2)^2}}{\sqrt{c(x_1 - x_2)^2 + (y_1 + y_2)^2} - \sqrt{c(x_1 - x_2)^2 + (y_1 - y_2)^2}}.$$

Table 4 shows that the proposed isometries well-preserve the distances.

Table 4: Validation of the proposed isometries between the hyperbolic models.

| $c$ | $T_{\mathcal{P}_c \to \mathcal{L}_c}$ | $T_{\mathcal{G}_c \to \mathcal{P}_c}$ | $T_{\mathcal{G}_c \to \mathcal{L}_c}$ |
|------|------|------|------|
| 0.25 | 5.72e−13 | 1.50e−13 | 4.76e−13 |
| 0.50 | 8.41e−13 | 3.10e−13 | 6.23e−13 |
| 1.00 | 7.84e−13 | 4.22e−13 | 4.88e−13 |
| 1.50 | 3.22e−12 | 2.51e−12 | 8.30e−13 |
| 2.00 | 1.89e−12 | 1.20e−12 | 8.86e−13 |

## C.2 THE LORENTZ MODEL OPERATIONS

The $n$-dimensional Lorentz model with curvature $-c$ is $\mathcal{L}_c^n$ where the manifold is $\{\mathbf{x} \in \mathbb{R}^{n+1} \mid \langle \mathbf{x}, \mathbf{x} \rangle_{\mathcal{L}_c} = -\frac{1}{c}\}$, where $\langle \mathbf{x}, \mathbf{y} \rangle_{\mathcal{L}_c}$ is the Lorentzian product computed as $\langle \mathbf{x}, \mathbf{y} \rangle_{\mathcal{L}_c} = -\mathbf{x}_0 \mathbf{y}_0 + \sum_{i=1}^n \mathbf{x}_i \mathbf{y}_i$. The exponential map of the Lorentz model is defined as:

$$\exp_{\mathbf{x}}^c(\mathbf{v}) = \cosh(\alpha)\mathbf{x} + \sinh(\alpha)\frac{\mathbf{v}}{\alpha}, \tag{26}$$

and the log map of the Lorentz model is defined as:

$$\log_{\mathbf{x}}^c(\mathbf{y}) = \frac{\cosh^{-1}(\beta)}{\sqrt{\beta^2 - 1}}(\mathbf{y} - \beta\mathbf{x}), \tag{27}$$

where $\alpha = \sqrt{c\langle \mathbf{v}, \mathbf{v} \rangle_{\mathbb{L}_c}}$ and $\beta = -c\langle \mathbf{x}, \mathbf{y} \rangle_{\mathbb{L}_c}$.

## D IMPLEMENTATION DETAILS

In this section, we introduce the implementation details for the density estimation experiment.

For the encoder and the decoder of binarized-MNIST and binarized-Omniglot, we use a two-layer fully connected neural network, where the dimension of the hidden units is 200 with the hyperbolic tangent activation, following the setting used in the importance weighted VAE (). For binarized-Breakout, the encoder is a four-layer convolutional neural network with leaky ReLU activation followed by a fully connected layer. The decoder consists of a fully connected layer followed by three transposed convolutional layers with ReLU activation. For both the encoder and decoder used in the binarized-Breakout, we place the batch normalization layer at the end of all layers. Since the datasets are binarized, we use the Bernoulli distribution as the output of decoders for all tasks.

For training, we use Adam optimizer (Kingma & Ba, 2014) with a constant learning rate 1e−3 and set the batch size to 100. We train the VAEs for 300 epochs on binarized-MNIST and binarized-Omniglot and 200 epochs on binarized-Breakout. The trained VAEs are evaluated by the log-likelihood on the test set for each task with importance-weighted sampling (Burda et al., 2015). The models learned from binarized-MNIST and binarized-Omniglot use 500 samples for the importance weighted sampling and the models learned from binarized-Breakout use 50 samples for the importance weighted sampling.

# E    NUMERICAL STABILITY

We conduct an analysis of the numerical stability of the PGM normal distribution compared to the HWN and Poincaré normal. During the density estimation experiment, the HWN VAE and Poincaré VAE are often shown to be numerically unstable and fail to run in binarized-Breakout. Similar observations have been reported in several previous works (Mathieu et al., 2019; Chen et al., 2021; Skopek et al., 2019).

The hyperbolic wrapped normal uses the exponential map when transforming the output of the encoder to the Lorentz model and during the sampling, as described in Equation 3. The overlapped Lorentz model exponential map often causes an overflow due to the hyperbolic functions in the exponential map such as $\cosh$ and $\sinh$. Note that the hyperbolic functions exponentially grow with the positive input value.

In the training of Poincaré VAE, the KL divergence between the variational distribution and the prior distribution needs to be approximated by the log-probability of the samples due to the absence of closed-form KL divergence in Poincaré normal. To compute the log probability of a given sample, the distance between two Poincaré disk model points, the sample and the Fréchet mean of the distribution needs to be calculated, where the distance function of the Poincaré disk model is defined as:

$$d_{\mathcal{P}}^c(\mathbf{x}, \mathbf{y}) = \frac{1}{\sqrt{c}} \cosh^{-1}\left(1 + 2c \frac{\|\mathbf{x} - \mathbf{y}\|^2}{(1 - c\|\mathbf{x}\|^2)(1 - c\|\mathbf{y}\|^2)}\right), \tag{28}$$

where $\|\cdot\|$ is the Euclidean norm. The denominator term is unstable when $\|\mathbf{x}\|$ or $\|\mathbf{y}\|$ is close to value $\frac{1}{\sqrt{c}}$, which is occured when $\mathbf{x}$ and $\mathbf{y}$ are near the border of the Poincaré disk.

PGM-normal, on the other hand, the KL divergence between an arbitrary PGM normal and $\mathcal{K}_c(\mathbf{0}, I, I)$, which is the only operation used during the training of GM-VAE, can be stably computed using the log-covariance. For example, the KL divergence between an univariate Gaussian distribution $\mathcal{N}(\mu, \sigma)$ and the prior distribution mentioned above written as Equation 14 can be computed with $\log \sigma^2$. The KL divergence between two Gamma distributions, $\mathrm{Gamma}(a_1, b_1)$ and $\mathrm{Gamma}(a_2, b_2)$, written as:

$$D_{\mathrm{KL}}(\mathrm{Gamma}(a_1, b_1) \parallel \mathrm{Gamma}(a_1, b_1)) = a_2 \log \frac{b_1}{b_2} - \ln \frac{\Gamma(a_1)}{\Gamma(a_2)} + (a_1 - a_2)\psi(a_1) - (1 - \frac{b_2}{b_1})a_1, \tag{29}$$

where $\psi$ is the digamma function, can be stablly computed using $\log b_1$ when $b_1$ is large due to small $\beta$ and $\gamma$ in the factorization Equation 6.

# F    LATENT SPACE ANALYSIS

In this section, we provide additional visualizations of the learned representations of GM-VAE. Figure 4 shows the t-SNE visualization of the Euclidean VAE learned on binarized-MNIST. Figure 5 shows the interpolation results of the binarized-MNIST representations along the $\mu$ axis, by changing the value of $\alpha$ on three dimensions and fixing all the other values including $\beta$. Figure 7 and Figure 6 show the latent representations from all the dimensions with binarized-MNIST and binarized-Breakout respectively.

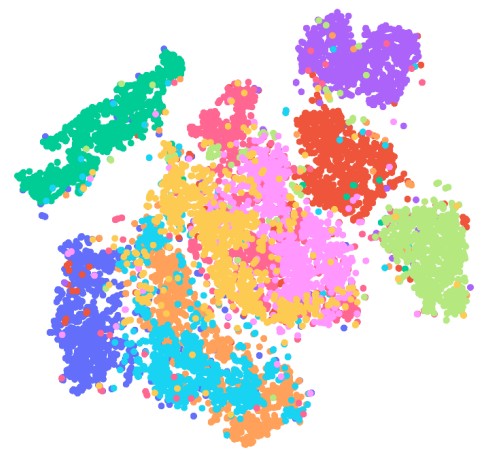

Figure 4: t-SNE visualization with representations learned from Euclidean-VAE.

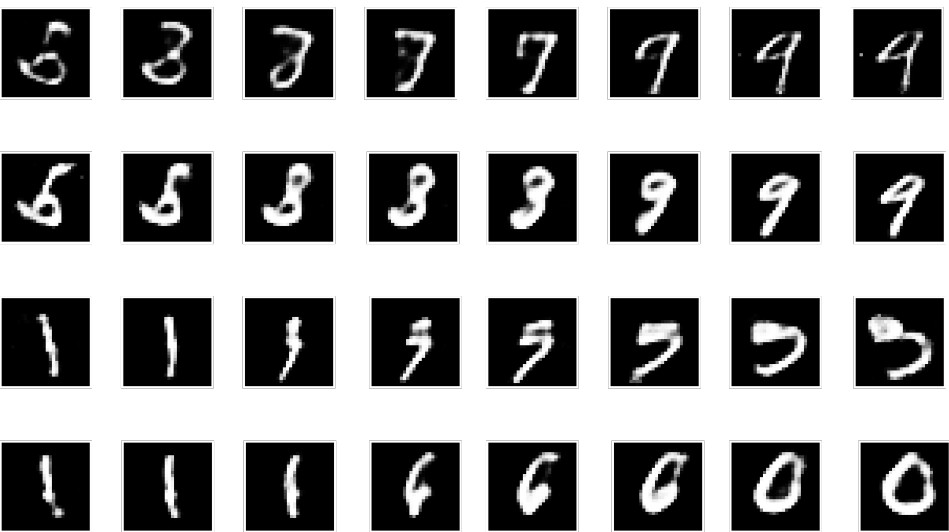

Figure 5: Interpolation results of the representations learned from GM-VAE along the $\mu$ axis.

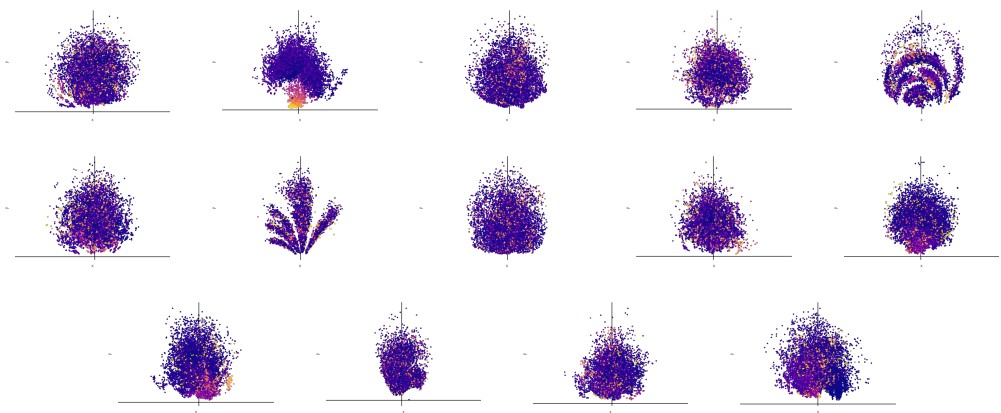

Figure 6: The binarized-Breakout representations learned from GM-VAE.

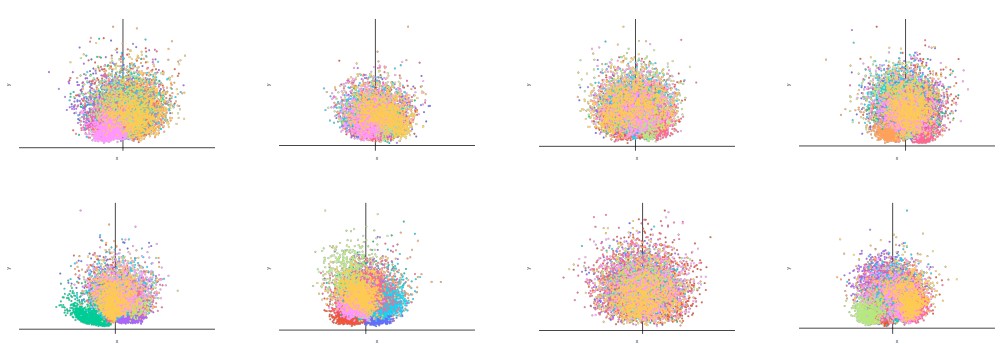

Figure 7: The binarized-MNIST representations learned from GM-VAE.

