# OpenReview forum: "GM-VAE: Representation Learning with VAE on Gaussian Manifold"
_ICLR.cc/2023/Conference — Submitted to ICLR 2023_

### Official Review · Reviewer_R7Dy · 2022-10-25

**Confidence:** 3
**Clarity, Quality, Novelty And Reproducibility:** See above.
**Correctness:** 3
**Technical Novelty And Significance:** 1
**Empirical Novelty And Significance:** 1
**Recommendation:** 3

**Strength And Weaknesses:**

Weakness:

(1) The hyperparameter $c$ only changes the scaling of $\mu$ and $\alpha$. It does not influence the representation capability of the VAE, and can be absorbed into the encoder/decoder model. The motivation that "generalization performances of hyperbolic VAEs can be improved with varying curvatures" is weak. ---- by introducing hyperparameters, one always have room for improving the generalization performance.

(2) Although well-motivated from manifold theory, the distribution over the manifold $\mathbb{R} \times \mathbb{R}_+$  reduces to a Normal distribution and a Gamma distribution. Instead, the properties of the manifolds are only used in the exponential map of the Lorentz model and the isometry between Lorentz space and the Gaussian manifold, the former being a known result in the previous paper.

(3)  It is not well-motivated to model the latent space as a Gaussian distribution. The connection of the proposed VAE to hierarchical VAE is weak.

(4) The experiments. From Table 2 I cannot see any benefit of the proposed method. It only works on Breakout dataset. Besides, why do you need to do binarization instead of modelling the original data?

**Summary Of The Paper:**


The paper defines PGM-normal, a distribution over the family of diagonal Gaussian distributions, which forms a manifold. With this distribution, they propose a variant of VAE whose latent space is this manifold. They propose two transformations for the VAE to map between Euclidean space and the manifold. Empirically, they show that their method matches the performance of existing hyperbolic VAEs.

**Summary Of The Review:**

The theoretical part is largely known. The motivation of the paper is weak. The experiments only show the proposed method works.

---

> ### Author Response · Authors · 2022-11-18
> **Response to Reviewer R7Dy (2/2)**
>
> > The connection of the proposed VAE to hierarchical VAE is weak.
>
> The latent space of GM-VAE is a set of diagonal Gaussian distributions,  each of which can define a distribution of the latent space with a standard VAE. Although we haven’t used the sampled variables of GM-VAE as parameters of Gaussian distribution, conceptually, we can consider them as Gassuian. Note that here the hierarchical VAE does not indicate the explicit hierarchical VAE model proposed in [5], but we use the term as a model with hierarchical priors (priors over priors) from a general Bayesian perspective.
>
> > The experiments. From Table 2 I cannot see any benefit of the proposed method. It only works on Breakout dataset.
>
> In practice, Poincar\’e VAE has many numerical issues, including slow sampling and numerical stability issues. The proposed GM-VAE, however, provides a hyperbolic VAE that overcomes practical issues by using a statistical distance on the hyperbolic space (or the Gaussian manifold), while preserving competitive performances.
>
> GM-VAE results 6.60s/epoch in training time which is faster than P-VAE about 2.68x (The results are from MNIST w/ latent dimension 20). GM-VAE shows competitive results with P-VAE among three out of six settings in MNIST and Omniglot. Furthermore, GM-VAE outperforms the hyperbolic VAE with HWN, which is the standard choice of hyperbolic VAE in practice [2,3,4], in almost all the settings. This reveals that GM-VAE is a better alternative of the previous hyperbolic VAE in practice.
>
> > Besides, why do you need to do binarization instead of modelling the original data?
>
> Binarizing images is a common technique used in many previous work [1,3,6,7]. This enables training generative model, especially VAE, easier without harming the semantic in the original images. The binarization is not necessity, as shown in an additional experiment with CIFAR-10.
>
> **References**
>
> [1] Emile Mathieu, Charline Le Lan, Chris J. Maddison, Ryota Tomioka, and Yee Whye Teh. Continuous hierarchical representations with poincaré variational auto-encoders. In H. Wallach, H. Larochelle, A. Beygelzimer, F. d'Alché-Buc, E. Fox, and R. Garnett, editors, Advances in Neural Information Processing Systems. Curran Associates, Inc., 2019.
>
> [2] Emile Mathieu and Maximilian Nickel. Riemannian continuous normalizing flows. Advances in Neural Information Processing Systems, 2020.
>
> [3] Skopek, Ondrej, Octavian-Eugen Ganea, and Gary Bécigneul. "Mixed-curvature variational autoencoders." arXiv preprint arXiv:1911.08411 (2019).
>
> [4] Joey Bose, Ariella Smofsky, Renjie Liao, Prakash Panangaden, and Will Hamilton. Latent variable modelling with hyperbolic normalizing flows. In International Conference on Machine Learning, 2020.
>
> [5] Child, Rewon. "Very deep vaes generalize autoregressive models and can outperform them on images." arXiv preprint arXiv:2011.10650 (2020).
>
> [6] Burda, Yuri, Roger Grosse, and Ruslan Salakhutdinov. "Importance weighted autoencoders." arXiv preprint arXiv:1509.00519 (2015).
>
> [7] Davidson, Tim R., et al. "Hyperspherical variational auto-encoders." arXiv preprint arXiv:1804.00891 (2018).

---

> ### Author Response · Authors · 2022-11-18
> **Response to Reviewer R7Dy (1/2)**
>
> We sincerely thank you for the careful and insightful comments. We address the questions below.
>
> > It is not well-motivated to model the latent space as a Gaussian distribution.
>
> The motivation of the research is to relieve the practical issues in the existing hyperbolic VAEs by using the Gaussian manifold. Although Poincar\’e VAE shows better performance than the other hyperbolic VAEs in previous work [1][3], it is not the standard choice of hyperbolic VAE in practice due to slow sampling and numerical stability issues [2][3][4].
>
> To overcome the practical issues, we focus on a hyperbolic space where each point is a pair of two parameters of a Gaussian distribution. Focusing on the analytic form of KL divergence between two points (i.e., two Gaussian distributions), we further analyze the KL divergence as a statistical distance. To deeply utilize the connection between hyperbolic space and the set of Gaussian distributions, we show that the local approximating property of the KL divergence remains even in the Riemannian manifold with an arbitrary negative constant curvature. This work, thus, provides the theoretical background on developing the proposed pseudo normal distribution on the Gaussian manifold with an arbitrary negative constant curvature. In other words, we present a novel method of using the analytical properties, geometrical properties (e.g., metric tensors), and probabilistic semantic lying on the Gaussian manifold viewing it as hyperbolic space, which indeed helps overcoming the practical issues of existing hyperbolic VAEs.
>
> > The hyperparameter c only changes the scaling of $\mu$ and $\alpha$. It does not influence the representation capability of the VAE, and can be absorbed into the encoder/decoder model. The motivation that "generalization performances of hyperbolic VAEs can be improved with varying curvatures" is weak. ---- by introducing hyperparameters, one always have room for improving the generalization performance.
>
> Of course, from a model parameterization perspective, the curvature parameter $c$ only changes the scaling of $\mu$ and $\alpha$ as you mentioned. However, from geometric perspectives, this induces the changes in underlying geometry, hence it changes the metric tensor of the latent space.  This then affects the geometrical properties of PGM normal which utilizes the statistical distance related to the metric tensor. This is definitely different from just letting the parameters be freely learnable. Some studies on the representations with varying curvature show that the curvature affects the representation capability of the VAE [1,3]. The curvature $c$ may indeed be absorbed in the encoder/decoder model. Explicitly modeling $c$, however, is inevitable in setting the latent space exactly to hyperbolic space with varying curvature.
>
> > Although well-motivated from manifold theory, the distribution over the manifold $\mathbb{R} \times \mathbb{R}_+$ reduces to a Normal distribution and a Gamma distribution. Instead, the properties of the manifolds are only used in the exponential map of the Lorentz model and the isometry between Lorentz space and the Gaussian manifold, the former being a known result in the previous paper.
>
> The derivation of PGM normal comes from the statistical distance that locally approximates the squared distance, which is one of the most important geometrical properties of the Gaussian manifold. In this sense, the geometrical property of the Gaussian manifold is well implied in PGM normal, and it can be even nicely factorizable. The factorization does not harm any geometrical property.
>
> Although using the exponential map of the Lorentz model is a known method, it can not be directly applied to the Gaussian manifold without the isometry between the Gaussian manifold and the Lorentz model. This is the reason why we propose the isometry between the Gaussian manifold and the Lorentz model with an arbitrary curvature, in Appendix C. This is the first time to propose such isometry, while the previous work focuses on isometries with only a curvature value of -1.

---

### Official Review · Reviewer_k6FC · 2022-10-27

**Confidence:** 2
**Correctness:** 2
**Technical Novelty And Significance:** 2
**Empirical Novelty And Significance:** 2
**Recommendation:** 3

**Clarity, Quality, Novelty And Reproducibility:**

The purpose or geometric meaning of the method is not clear.
The idea of this work is somewhat novel.


**Strength And Weaknesses:**

Strength：

The idea to learn a new manifold for data is interesting.


Weaknesses:

1. The motivation of the methods in the manuscript is not clear and most hypotheses lack explanation. For example, why use diagnal Gaussian distributions.

2. The authors only introduce different methods by plain descriptions and isolated formulas, and does not analyze or summarize the characteristics of different methods in manifold learning. For example, what scenario does the Riemannian Gaussian distribution apply to, and what scenario does the wrapped normal distribution apply to？

3. The authors directly give the distribution expression of PGM in (5) without explaining its geometric meaning.

4. How to get the  conclusion of ”With this aspect, GM-VAE can be considered as a hierarchical VAE with an additional prior over the Gaussian prior“？

5. The experimental results are not convincing. In Table 2，the proposed method does not perform as well as P-VAE and the authors do not make any anlysis between P-VAE and GM-VAE.

6. For the results of visualization of latent space such as in Figure 2 and Figure 3, how to evaluate them?

7. What's the difference between the proposed GM-VAE and some Gaussian-process-based manifold learning methods, such as GPLVM or VAE-DGP?


**Summary Of The Paper:**

In this paper, a pseudo Gaussian manifold normal distribution is proposed as the prior of latent representation of data, and the latent representation of data is learned in the framework of VAE, so as to realize a Gaussian manifold learning.

**Summary Of The Review:**

The manuscript presents an interesting idea, but the details and rationality of the hypothesis are not clear, and the experimental results are difficult to support and explain the hypothesis.

---

> ### Author Response · Authors · 2022-11-18
> **Response to Reviewer k6FC (2/2)**
>
> > The experimental results are not convincing. In Table 2, the proposed method does not perform as well as P-VAE and the authors do not make any analysis between P-VAE and GM-VAE.
>
> We first note that the representation power of GM-VAE and Poincare VAE are the same since they are defined over the same hyperbolic latent space.
>
> In practice, P-VAE has many numerical issues, including slow sampling and numerical stability issues. The proposed GM-VAE, however, provides a hyperbolic VAE that overcomes practical issues by using a statistical distance on the hyperbolic space (or the Gaussian manifold), while preserving competitive performances.
>
> GM-VAE results 6.60s/epoch in training time which is faster than P-VAE about 2.68x (The results are from MNIST w/ latent dimension 20). In the aspect of numerical stability, GM-VAE is the only hyperbolic VAE which is successfully trained with Breakout images. GM-VAE shows competitive results with P-VAE among three out of six settings in MNIST and Omniglot. GM-VAE shows better performance than L-VAE, the previous standard choice of hyperbolic VAEs in practice. As a result, GM-VAE shows better stability, lower computational complexity, and higher geometrical interpretability (as shown in the analysis from Sec. 5.2). We will further update these descriptions in the revised version.
>
> > For the results of visualization of latent space such as in Figure 2 and Figure 3, how to evaluate them?
>
> In Figure 3, we compute the Pearson correlation between the \beta value obtained from the encoder and the cumulative reward of the Breakout image. The resulting correlation is 0.655.
>
> > What's the difference between the proposed GM-VAE and some Gaussian-process-based manifold learning methods, such as GPLVM or VAE-DGP?
>
> GPLVM learns the encoder and decoder from the Gaussian process prior. Our method focuses on the latent space of the learned representation. VAE-DGP sets the latent variable as a time-series sequence and utilizes the Gaussian process to model the sequences. Our method set the latent variable to be a diagonal Gaussian distribution.
>
> **References**
>
> [1] Emile Mathieu, Charline Le Lan, Chris J. Maddison, Ryota Tomioka, and Yee Whye Teh. Continuous hierarchical representations with poincaré variational auto-encoders. In H. Wallach, H. Larochelle, A. Beygelzimer, F. d'Alché-Buc, E. Fox, and R. Garnett, editors, Advances in Neural Information Processing Systems. Curran Associates, Inc., 2019.
>
> [2] Emile Mathieu and Maximilian Nickel. Riemannian continuous normalizing flows. Advances in Neural Information Processing Systems, 2020.
>
> [3] Skopek, Ondrej, Octavian-Eugen Ganea, and Gary Bécigneul. "Mixed-curvature variational autoencoders." arXiv preprint arXiv:1911.08411 (2019).
>
> [4] Joey Bose, Ariella Smofsky, Renjie Liao, Prakash Panangaden, and Will Hamilton. Latent variable modelling with hyperbolic normalizing flows. In International Conference on Machine Learning, 2020.
>
> [5] Maximillian Nickel and Douwe Kiela. Learning continuous hierarchies in the Lorentz model of hyperbolic geometry. In Jennifer Dy and Andreas Krause, editors, Proceedings of the 35th International Conference on Machine Learning, Proceedings of Machine Learning Research. PMLR, 2018.
>
> [6] Child, Rewon. "Very deep vaes generalize autoregressive models and can outperform them on images." arXiv preprint arXiv:2011.10650 (2020).

---

> ### Author Response · Authors · 2022-11-18
> **Response to Reviewer k6FC (1/2)**
>
> We acknowledge your help for improving our paper with careful reviews and comments. We address the specific comments and concerns in detail below.
>
> > The motivation of the methods in the manuscript is not clear and most hypotheses lack explanation. For example, why use diagonal Gaussian distributions.
>
> We use the diagonal Gaussian manifold because it is a product of two-dimensional hyperbolic spaces. Here, we describe the motivation of the product of two-dimensional hyperbolic spaces.
>
> The motivation of the research is to relieve the practical issues in the existing hyperbolic VAEs by using the diagonal Gaussian manifold. Although Poincar\’e VAE shows better performance than the other hyperbolic VAEs in previous work [1,3], it is not the standard choice of hyperbolic VAE in practice due to slow sampling and numerical stability issues [2,3,4].
>
> To overcome the practical issues, we focus on the hyperbolic space where each point is the parameters of a diagonal Gaussian distribution. Focusing on the analytic form of KL divergence between two points, we further analyze the KL divergence as a statistical distance. To fully leverage connection between hyperbolic space and the Gaussian manifold, we extend the local approximating property of the KL divergence so it can also be used in the Reimannian manifold with an arbitrary negative constant curvature. This work, thus, enables developing a pseudo normal distribution on Gaussian manifold an arbitrary negative constant curvature, i.e., provides a method of using the analytical properties, geometrical properties, and even probabilistic semantic lying on the Gaussian manifold viewing it as the hyperbolic space.
>
> > The authors only introduce different methods by plain descriptions and isolated formulas, and does not analyze or summarize the characteristics of different methods in manifold learning. For example, what scenario does the Riemannian Gaussian distribution apply to, and what scenario does the wrapped normal distribution apply to?
>
> Poincar\’e normal, the Riemannian normal of hyperbolic space, often shows better performance than the hyperbolic wrapped normal distribution [1,3]. However, the Poincar\’e normal suffers from several practical issues, including numerical stability and high computational complexity in sampling, due to the openness of the Poincar\’e disk model [5]. The hyperbolic wrapped normal distribution, which is the standard choice of hyperbolic VAEs in practice [2,3,4], is introduced by using the exponential map of hyperbolic space, but lacks geometrical interpretability and suffers from worse performance (even compared to our method).
>
> > The authors directly give the distribution expression of PGM in (5) without explaining its geometric meaning.
>
> We explain the geometric meaning of the PGM normal in section 3.1. Here, we reiterate the geometric meaning of PGM normal.
> We exploit the KL-divergence, a statistical distance, which we show a local approximation of the squared Fisher-Rao distance, shown in Eq. 4. This explains that PGM normal uses an approximated Mahalanobis distance, where the usual hyperbolic (or Riemannian) normal is derived from.
>
> > How to get the conclusion of “With this aspect, GM-VAE can be considered as a hierarchical VAE with an additional prior over the Gaussian prior”？
>
> The latent space of GM-VAE is a set of diagonal Gaussian distributions,  each of which can define a distribution of the latent space with a standard VAE. Although we haven’t used the sampled variables of GM-VAE as parameters of Gaussian distribution, conceptually, we can consider them as Gassuian. Note that here the hierarchical VAE does not indicate the explicit hierarchical VAE model proposed in [6], but we use the term as a model with hierarchical priors (priors over priors) from  a general Bayesian perspective.

---

### Official Review · Reviewer_G6cG · 2022-11-04

**Confidence:** 3
**Clarity, Quality, Novelty And Reproducibility:** 1. Clarity
**Correctness:** 3
**Technical Novelty And Significance:** 2
**Empirical Novelty And Significance:** 2
**Recommendation:** 3

**Strength And Weaknesses:**

Strengths

1. The proposed method is novel, interesting and makes use of clever engineering details that are rather intuitive.
2. Empirical results clearly indicate that the proposed method is superior in terms of numerical stability to existing hyperbolic latent space methods, over certain datasets (such as binarized-Breakout). In fact, in my opinion, the main selling point of the method should be in terms of numerical stability and the authors may consider bringing the discussion in Appendix E into the main text.
3. Empirical results in terms of negative test likelihood do not suffer, compared to other methods.
4. The introduction to hyperbolic latent space is comprehensive and self-contained. Overall I find the language clear and the flow easy to follow, despite small typos.

Weaknesses

1. Some notations are not clearly defined. In figure 1, the calligraphic E space (I suspect is Euclidean space) is not defined and the calligraphic L^2 space is only defined in passing in subsection 3.3. I suggest that the authors at least make the notation in the figures and its caption self-contained.  What does the triangle in Table 1 mean? How is it different from a circle?
2. Although a lot is mentioned about information geometry and the incorporation of closed form KL, the paper does not clearly mention (or hint at) specific mathematical guarantees for the proposed model. As a result, a lot of the justifications for the proposed method are heuristical and do not form concrete theoretical backings.
3. Empirical results are rather limited in quantity. There are only a total of 3 datasets and the authors only demonstrate better stability in 1 of these 3 datasets. The proposed methods also do not seem to scale as well as competing methods when increasing the latent space dimension.


**Summary Of The Paper:**

The paper introduces and studies a new type of hyperbolic latent space called Gaussian manifold in the context of variational auto-encoder in order to tackle numerical and sampling issues with existing hyperbolic latent spaces. To make this latent space works in practice, the authors designed a geometric transformation at the last encoder step and the first decoder step; as well as a novel distribution over Gaussian manifold called a pseudo Gaussian manifold normal distribution that makes use of information geometry, easy to sample from, and easy to compute KL divergence. Finally, comprehensive testing on empirical datasets are carried out to highlight the effectiveness of the scheme.






**Summary Of The Review:**

Despite the paper containing some interesting and novel ideas on how to make the Gaussian manifold suitable for VAE setting, I find that the theoretical motivation for said ideas is heuristical at best and there is not much empirical evidence to demonstrate the strength of the proposed method. Therefore, I recommend rejection.

---

> ### Author Response · Authors · 2022-11-18
> **Response to Reviewer G6cG**
>
> We appreciate your constructive feedback. We will add the missing definitions in the revised manuscript. Here, we address the your questions mentioned.
>
> > What does the triangle in Table 1 mean? How is it different from a circle?
>
> Both symbols imply that each method utilizes easy sampling methods. The circle symbol means it is easy without sacrificing computational complexity. The triangle symbol means that the method is computationally expensive.
>
> > Although a lot is mentioned about information geometry and the incorporation of closed form KL, the paper does not clearly mention (or hint at) specific mathematical guarantees for the proposed model. As a result, a lot of the justifications for the proposed method are heuristical and do not form concrete theoretical backings.
>
> The mathematical guarantees of our model come from the geometrical property of hyperbolic space. It is known that any tree can be embedded in two-dimensional hyperbolic space with arbitrary distortion, while Euclidean space can’t [1]. This enables hyperbolic space to be an efficient medium in modeling hierarchical structure. Based on the theoretical background, we propose a new method of using hyperbolic space as the latent space for VAE, where each point in the space is set to be the parameters of a diagonal Gaussian distribution. Our experiment shows competitive results with the commonly-used hyperbolic VAEs in density estimation (with better numerical stability) while showing a well-preserved hierarchical structure (e.g., in Breakout experiment).
>
> > Empirical results are rather limited in quantity. There are only a total of 3 datasets and the authors only demonstrate better stability in 1 of these 3 datasets. The proposed methods also do not seem to scale as well as competing methods when increasing the latent space dimension.
>
> We first note that the representation power of GM-VAE and Poincar\’e VAE are identical since they are defined over the same hyperbolic latent space.
>
> GM-VAE has more advantages which are uncovered by the quantities. Although Poincar\’e VAE shows better performance than the other hyperbolic VAEs in previous work [2,4], it is not the standard choice of hyperbolic VAE in practice due to slow sampling and numerical stability issues [3,4,5]. For example, GM-VAE results in 6.60s/epoch in training time which is faster than Poincar\’e VAE about 2.68x (Estimated from MNIST with latent dimension 20).
>
> The proposed GM-VAE not only overcomes practical issues but also shows competitive performances when the models succeed to be trained. GM-VAE shows competitive results with Poincar\’e VAE among three out of six settings in MNIST and Omniglot. Furthermore, GM-VAE outperforms the hyperbolic VAE with HWN, which is the standard choice of hyperbolic VAE in practice, in almost all the settings. This reveals that GM-VAE is a better alternative of the previous hyperbolic VAEs in practice.
>
> In the aspect of numerical stability, considering the high dimensionality of Breakout images, we thought that the numerical stability of GM-VAE compared to Poincar\’e VAE is well described. Appreciating the suggestion, we will further show more settings that emphasize numerical stability of our method in the future, i.e.., using higher dimensional image datasets.
>
> **References**
>
> [1] N. Linial, E. London, and Y. Rabinovich. The geometry of graphs and some of its algorithmic applications. In Proceedings 35th Annual Symposium on Foundations of Computer Science, 1994.
>
> [2] Emile Mathieu, Charline Le Lan, Chris J. Maddison, Ryota Tomioka, and Yee Whye Teh. Continuous hierarchical representations with poincaré variational auto-encoders. In H. Wallach, H. Larochelle, A. Beygelzimer, F. d'Alché-Buc, E. Fox, and R. Garnett, editors, Advances in Neural Information Processing Systems. Curran Associates, Inc., 2019.
>
> [3] Emile Mathieu and Maximilian Nickel. Riemannian continuous normalizing flows. Advances in Neural Information Processing Systems, 2020.
>
> [4] Skopek, Ondrej, Octavian-Eugen Ganea, and Gary Bécigneul. "Mixed-curvature variational autoencoders." arXiv preprint arXiv:1911.08411 (2019).
>
> [5] Joey Bose, Ariella Smofsky, Renjie Liao, Prakash Panangaden, and Will Hamilton. Latent variable modelling with hyperbolic normalizing flows. In International Conference on Machine Learning, 2020.

---

### Author Response · Authors · 2022-11-18
**Special thanks to the reviewers.**

We thanks for the comments that reveal the weaknesses of our paper. For each reviewer, we responded to the corresponding review. Additional feedbacks and discussion are always welcome!

---

### Decision · Program_Chairs · 2023-01-20

**Decision:**

Reject

**Justification For Why Not Higher Score:**

The motivation of the methods in the manuscript is not clear and most hypotheses lack explanation. Empirical results are rather limited in quantity and not convincing. It is not well-motivated to model the latent space as a Gaussian distribution. The connection of the proposed VAE to hierarchical VAE is weak.




**Justification For Why Not Lower Score:**

N/A

**Metareview: Summary, Strengths And Weaknesses:**

The paper introduces and studies a new type of hyperbolic latent space called Gaussian manifold in the context of variational auto-encoder in order to tackle numerical and sampling issues with existing hyperbolic latent spaces. To make this latent space works in practice, the authors designed a geometric transformation at the last encoder step and the first decoder step; as well as a novel distribution over Gaussian manifold called a pseudo Gaussian manifold normal distribution that makes use of information geometry, easy to sample from, and easy to compute KL divergence. Finally, comprehensive testing on empirical datasets are carried out to highlight the effectiveness of the scheme.

The proposed method is novel, interesting and makes use of clever engineering details that are rather intuitive. However, the motivation of the methods in the manuscript is not clear and most hypotheses lack explanation. Empirical results are rather limited in quantity and not convincing. It is not well-motivated to model the latent space as a Gaussian distribution. The connection of the proposed VAE to hierarchical VAE is weak.